# The Density of Group I mGlu_5_ Receptors Is Reduced along the Neuronal Surface of Hippocampal Cells in a Mouse Model of Alzheimer’s Disease

**DOI:** 10.3390/ijms22115867

**Published:** 2021-05-30

**Authors:** Alejandro Martín-Belmonte, Carolina Aguado, Rocío Alfaro-Ruiz, José Luis Albasanz, Mairena Martín, Ana Esther Moreno-Martínez, Yugo Fukazawa, Rafael Luján

**Affiliations:** 1Synaptic Structure Laboratory, Instituto de Investigación en Discapacidades Neurológicas (IDINE), Departamento de Ciencias Médicas, Facultad de Medicina, Campus Biosanitario, Universidad de Castilla-La Mancha, C/Almansa 14, 02006 Albacete, Spain; Alejandro.Martin@uclm.es (A.M.-B.); Carolina.Aguado@uclm.es (C.A.); Rocio.Alfaro@uclm.es (R.A.-R.); AnaEsther.Moreno@uclm.es (A.E.M.-M.); 2Regional Center of Biomedical Research (CRIB), Department of Inorganic, Organic and Biochemistry, Faculty of Chemical and Technological Sciences, School of Medicine of Ciudad Real, Universidad de Castilla-La Mancha, 13071 Ciudad Real, Spain; Jose.Albasanz@uclm.es (J.L.A.); Mairena.Martin@uclm.es (M.M.); 3Research Center for Child Mental Development, Life Science Innovation Center, Division of Brain Structure and Function, School of Medical Science, University of Fukui, Fukui 910-1193, Japan; yugo@u-fukui.ac.jp

**Keywords:** Alzheimer’s disease, hippocampus, mGlu receptors, immunohistochemistry, electron microscopy, freeze-fracture, AD mouse model

## Abstract

Metabotropic glutamate receptor subtype 5 (mGlu_5_) is implicated in the pathophysiology of Alzheimer’s disease (AD). However, its alteration at the subcellular level in neurons is still unexplored. Here, we provide a quantitative description on the expression and localisation patterns of mGlu_5_ in the APP/PS1 model of AD at 12 months of age, combining immunoblots, histoblots and high-resolution immunoelectron microscopic approaches. Immunoblots revealed that the total amount of mGlu_5_ protein in the hippocampus, in addition to downstream molecules, i.e., G_q/11_ and PLCβ_1_, was similar in both APP/PS1 mice and age-matched wild type mice. Histoblots revealed that mGlu_5_ expression in the brain and its laminar expression in the hippocampus was also unaltered. However, the ultrastructural techniques of SDS-FRL and pre-embedding immunogold demonstrated that the subcellular localisation of mGlu_5_ was significantly reduced along the neuronal surface of hippocampal principal cells, including CA1 pyramidal cells and DG granule cells, in APP/PS1 mice at 12 months of age. The decrease in the surface localisation of mGlu_5_ was accompanied by an increase in its frequency at intracellular sites in the two neuronal populations. Together, these data demonstrate, for the first time, a loss of mGlu_5_ at the plasma membrane and accumulation at intracellular sites in different principal cells of the hippocampus in APP/PS1 mice, suggesting an alteration of the excitability and synaptic transmission that could contribute to the cognitive dysfunctions in this AD animal model. Further studies are required to elucidate the specificity of mGlu_5_-associated molecules and downstream signalling pathways in the progression of the pathology.

## 1. Introduction

Alzheimer’s disease (AD), the most common aetiology of dementia, is a progressive neurodegenerative condition characterised by a gradual decline in multiple cognitive functions and non-cognitive neuropsychiatric symptoms [1,2]. The three major neuropathology hallmarks of AD are the extracellular deposits of amyloid β (Aβ) peptides forming senile plaques, the intracellular accumulation of neurofibrillary tangles, which contain hyperphosphorylated Tau protein, and synapse loss [3,4]. These pathological alterations cause neuronal cell death in the hippocampus and cortical areas affecting memory formation, reasoning, language, and social behaviour [2]. Evidence suggests that Aβ and Tau oligomers contribute to dendritic spine and synaptic loss in AD, especially in the hippocampus [5], with subsequent disturbance of the glutamatergic neurotransmission machinery [6,7,8,9].

Glutamate is the main excitatory neurotransmitter in the hippocampus [10,11]. The effects of glutamate are mediated by activation of ionotropic and metabotropic receptors; both involved in the fine-tuning of neuronal responses and in neuronal plasticity, which underlies learning and memory formation [12,13]. Metabotropic glutamate (mGlu) receptors are coupled via G-proteins and second messengers to various effector mechanisms to mediate the slow component of excitatory responses [14]. To date, eight different subtypes of mGlu receptors have been identified (mGlu_1–8_), which have been classified into three groups (Group I, Group II and Group III) based on their sequence homology, transduction mechanism and pharmacology. Group I includes mGlu_l_ and mGlu_5_, which couple to the heterotrimeric G protein Gα_q/11_ and activate phospholipase C, resulting in increased inositol-1,4,5-trisphosphate formation and the release of Ca^2+^ from intracellular stores [15]. Activation of Group I mGlu receptors in the hippocampus leads to pyramidal cell excitability and transmission through the regulation of neurotransmitter receptors and ion channels [14,16]. Group I mGlu receptors are also important in the induction of long-term potentiation and long-term depression [17,18].

In situ hybridization and light microscopic immunohistochemical studies have shown that mGlu_5_ is widely expressed in the brain, being predominant in the hippocampus [19,20,21,22], where they play modulatory roles in regulating excitatory transmission at glutamatergic synapses and NMDA receptor-dependent synaptic plasticity [23]. An alteration in the signalling of mGlu_5_ is associated with neurological conditions and neurodegenerative diseases including AD [15,24]. It has been demonstrated that Aβ oligomer induces mGlu_5_ clustering, which leads to an elevation in intracellular calcium and excitatory synapse deterioration in AD model mice [25,26]. In addition, mGlu_5_ blockade reversed the learning and memory deficits observed in the AD mice [26]. However, it remains unclear how the distribution of mGlu_5_ is altered at the surface of hippocampal neurons in the AD mice.

To further understand the involvement of mGlu_5_ in the hippocampus of AD, we evaluated changes in their expression and distribution in the APP/PS1 model of AD using immunoblots, histoblots and high-resolution immunohistochemical techniques in combination with quantitative approaches. We provided evidence for a significant reduction in the surface expression of mGlu_5_ in CA1 pyramidal cells and DG granule cells.

## 2. Results

### 2.1. mGlu_5_ and Downstream Molecules in APP/PS1 Mice

Given that the transduction pathway mediating the response of mGlu_5_ involves the activation of PLC through the G protein G_q/11_, we first analysed, using immunoblots, the protein expression of mGlu_5_, PLCβ_1_ and G_q/11_ in hippocampal membranes from APP/PS1 and wild type mice (Figure 1A,B). As shown in Figure 1A, no significant differences were detected in mGlu_5_, PLCβ_1_ and G_q/11_ expression levels in APP/PS1 mice at 12 months of age compared to age-matched wild type mice (Figure 1A,B).

### 2.2. Similar Regional and Cellular Expression of mGlu_5_ in Control and APP/PS1 Mice

As immunoblots are not informative regarding potential regional- or cell-type-specific changes, we next analysed the region-dependent expression of mGlu_5_ in the brain of APP/PS1 and wild type mice using an mGlu_5_ subtype-specific antibody in conventional histoblots [27] at 12 months of age (Figure 1C–E). The brain expression of mGlu_5_ protein revealed marked region-specific differences, with the strongest immunolabelling in the hippocampus, caudate putamen and septum, moderate labelling in the cortex and thalamus, and very weak labelling in the cerebellum (Figure 1C,D). Quantification of mGlu_5_ expression levels revealed no differences between wild type and APP/PS1 mice (Figure 1E). Similar regional expression patterns and an absence of differences were also observed at 1 and 6 months of age (Appendix A).

We next analysed the laminar expression and distribution pattern of mGlu_5_ in the hippocampus using histoblots at 12 months of age (Figure 1F–H). Immunostaining for mGlu_5_ was widely expressed in all hippocampal subfields and dendritic layers of wild type and APP/PS1 mice, showing similar expression patterns in both genotypes (Figure 1F–H). Thus, in the CA1 region, mGlu_5_ expression was strong, with the *strata oriens* and *radiatum* showing the highest and the *stratum lacunosum-moleculare* showing lower expression levels (Figure 1F–H). The CA3 region and DG exhibited lower mGlu_5_ expression levels than the CA1 region, with the *strata lucidum* and *lacunosum-moleculare*, as well as the molecular layer and hilus of the DG, showing the lowest expression level within the hippocampus (Figure 1F–H). The *stratum pyramidale* and granule cell layer showed the weakest expression level throughout (Figure 1F–H). Similar laminar expression patterns were observed in the hippocampus of wild type and APP/PS1 mice at 1 and 6 months (Appendix A).

At the cellular level, although immunoreactivity for mGlu_5_ was present mainly in the neuropil in all dendritic layers, labelling could only be seen rarely outlining the somata and dendrites of scattered interneurons in all areas, particularly in the hilus, indicating the preferential distribution in CA1 and CA3 pyramidal cells and granule cells in the DG, with similar patterns in both APP/PS1 mice and WT (Appendix A).

### 2.3. Reduction of mGlu_5_ in the Surface of CA1 Pyramidal Cells in APP/PS1 Mice

Using the highly-sensitive SDS-FRL technique, we first investigated the organization of mGlu_5_ along the surface of CA1 pyramidal cells and their possible alteration in the hippocampus of APP/PS1 mice at 12 months of age. Electron microscopic analysis of the hippocampal replicas revealed immunogold labelling distributed only on P-faces of pyramidal cells, in agreement with the location of the sequence specifying the mGlu_5_ in the intracellular carboxy terminal tail. Consistent with previous pre-embedding immunogold labelling studies [21,22], mGlu_5_ was detected along the postsynaptic membrane of the entire somato-dendritic compartment of CA1 principal cells (Figure 2 and Figure 3).

In wild type mice, immunoparticles for mGlu_5_ were observed throughout the dendritic spines, dendritic shafts in *strata oriens*, *radiatum* and *lanunosum-moleculare* of the CA1 region (Figure 2A,C–F) and somata of pyramidal cells in the *stratum pyramidale* (Figure 2B). The neuronal compartments that showed the highest density of immunoparticles for mGlu_5_ were dendritic spines, and then dendritic shafts and somata (Figure 2A,C–F) (Table 1). In the eight neuronal compartments analysed, immunoparticles for mGlu_5_ were mostly found forming clusters (2847 immunoparticles out of 4170) along the neuronal surface (Figure 2A–F). Less frequently, mGlu_5_ immunoparticles were also distributed as a scattered pattern (1323 immunoparticles out of 4170) (Figure 2A–F). Immunoparticles for mGlu_5_ were only confined to somato-dendritic domains of CA1 pyramidal cells and not observed at pre-synaptic sites, which is consistent with previous studies [21,22]. E-faces of pyramidal cells are free of mGlu_5_ immunoparticles, as well as cross-fractures of dendrites or spines (Figure 2A–F). In APP/PS1 mice, immunoparticles for mGlu_5_ followed a similar subcellular localisation pattern to that described above for wild type, although the intensity of labelling changed significantly, with fewer immunoparticles per cluster and fewer clustered (399 immunoparticles out of 920) and scattered (521 immunoparticles out of 920) immunoparticles detected along the surface of CA1 pyramidal cells (Figure 2G–L).

Next, we performed a quantitative comparison of the mGlu_5_ densities in eight different somato-dendritic compartments in the CA1 region (Figure 3A) (Table 1). In wild type mice, our data revealed a graded increase in the density of mGlu_5_ immunoparticles from the soma to the dendritic spines in *strata radiatum* (Figure 3A). Although a somato-dendritic gradient was also observed in APP/PS1 mice, we found a significant decrease in mGlu_5_ densities in all layers analysed (Two-way ANOVA test and Bonferroni post hoc test, **** *p* < 0.0001) (Figure 3A) (Table 1). To determine how this reduction in density is taking place, we analysed the composition of clusters of mGlu_5_ immunoparticles along the neuronal surface in the eight compartments (Figure 3B), and kept the surface area and number of profiles similar to avoid neuronal alterations induced by AD (Table 2). Combining all oblique dendrites from *strata oriens*, *radiatum* and *lacunosum-moleculare*, we detected a total of 72 clusters with a range of 3 to 9 immunoparticles in APP/PS1 mice, while, for those with similar surface area, we detected 452 clusters with a range of 3 to 20 immunoparticles for mGlu_5_ in wild type mice (Figure 3B). Combining all spines from the three dendritic layers, we detected a total of 13 clusters with a range of 3 to 4 immunoparticles in APP/PS1 mice, while we detected 79 clusters with a range of 3 to 13 immunoparticles for mGlu_5_ in wild type mice (Figure 3B). The analysis performed individually in each of the eight compartments showed that the number and size of clusters of mGlu_5_ immunoparticles were reduced in the APP/PS1 mice compared to age-matched wild type (Table 2).

### 2.4. Reduction of mGlu_5_ in the Surface of DG Granule Cells in APP/PS1 Mice

Using the same methodological approaches, we also investigated the organisation of mGlu_5_ along the surface of granule cells in the DG (Figure 4). In wild type mice, mGlu_5_ immunoparticles were distributed in dendritic spines and shafts through the molecular layer of the DG, as well as in somata of granule cells in the granule cell layer, mainly forming clusters (689 immunoparticles out of 1192) but also scattered (503 immunoparticles out of 1192) along the neuronal surface (Figure 4A–C). In APP/PS1 mice, mGlu_5_ immunoparticles were observed in the same subcellular compartments, but immunolabelling intensity was reduced and the clustered distribution pattern (35 immunoparticles out of 211) observed in wild type changed to a predominantly scattered pattern (176 immunoparticles out of 211) (Figure 4D–F). This change in subcellular localisation pattern was confirmed performing quantitative analyses in three different somato-dendritic compartments in granule cells (Figure 4G–H). Thus, the graded increase in the density of mGlu_5_ immunoparticles from the soma to the dendritic spines observed in wild type mice was significantly reduced in all neuronal compartments of granule cells in APP/PS1 mice (Two-way ANOVA test and Bonferroni post hoc test, *** *p* < 0.001, **** *p* < 0.0001) (Figure 4G) (Table 1). This reduction in density was due to a reduction in the number and composition of mGlu_5_ immunoparticles forming clusters along the neuronal surface of granule cells (Figure 4H). Thus, in dendritic spines, we did not detect any cluster with three or more immunoparticles in APP/PS1 mice, while among those with similar surface area, we detected 26 clusters with a range of 3 to 16 immunoparticles for mGlu_5_ in wild type mice (Figure 4H). In oblique dendrites, we detected a total of eight clusters with a range of 3 to 5 immunoparticles in APP/PS1 mice, while we detected 75 clusters with a range of 3 to 20 immunoparticles for mGlu_5_ in wild type mice (Figure 4H).

### 2.5. Increase of mGlu_5_ in the Cytoplasm of Hippocampal Neurons in APP/PS1 Mice

The preceding data above show that the total amount of mGlu_5_ protein does not change in the hippocampus of APP/PS1, but the surface distribution in principal cells is reduced. Thus, we investigated the possible internalisation and accumulation of mGlu_5_ at intracellular sites of CA1 pyramidal cells and DG granule cells (Figure 5) of APP/PS1 mice at 12 months of age. For this purpose, we used the pre-embedding immunogold technique and quantitative analyses on tissue blocks taken from the middle part of the CA1 *stratum radiatum* and DG molecular layer. Similar subcellular localisation patterns were observed at 1 and 6 months of age (Appendix A).

At the ultrastructural level, immunoparticles for mGlu_5_ were detected along the extrasynaptic plasma membrane of dendritic spines and shafts of CA1 pyramidal cells and DG granule cells, as well as at intracellular sites associated with intracellular membranes in the same neuronal compartments, both in wild type and APP/PS1 mice (Figure 5A–D). Quantitative analysis of pre-embedding gold particles showed clear differences in the immunolabelling present in the plasma membrane versus intracellular sites in pyramidal cells (plasma membrane: 53.65% in wild type, *n* = 1281 particles, and 29.47% in APP/PS1, *n* = 707 particles; intracellular: 46.35% in wild type, *n* = 1115 particles, and 70.53% in APP/PS1, *n* = 1673 particles) (Figure 5E) and granule cells (plasma membrane: 48.60% in wild type, *n* = 1012 particles, and 32.56% in APP/PS1, *n* = 1081 particles; intracellular: 51.40% in wild type, *n* = 1054 particles, and 67.44% in APP/PS1, *n* = 2250 particles) (Figure 5F). These changes in subcellular localisation from the plasma membrane to intracellular sites were detected both in dendritic spines and dendritic shafts of CA1 pyramidal cells and DG granule cells (Figure 5E,F), thus demonstrating a redistribution of mGlu_5_ in hippocampal principal cells.

## 3. Discussion

G protein-coupled receptors (GPCRs) are involved in the pathogenesis of AD and in multiple stages of the processing of APP [28]. The mGlu_5_ subtype is a GPCRs critically involved in learning and memory [16,29]. Through several intracellular cascades, mGlu_5_ modulates neuronal excitability and synaptic plasticity, leading to LTP and LTD [30]. Compiling evidence highlights the potential therapeutic applicability of mGlu_5_ in AD pathophysiology [24,31,32,33]. The present study aimed at evaluating the possible changes in expression and localisation of mGlu_5_ in an animal model of AD at an advanced stage of the disease. In particular, we examined the subcellular localisation of mGlu_5_ in two hippocampal principal cells, the CA1 pyramidal cells and the DG granule cells in APP/PS1 mice of 12 months of age. Our findings demonstrate, for the first time, a reduction of mGlu_5_ in the neuronal surface of principal cells in the hippocampus of APP/PS1 mice. This decrease in mGlu_5_ all over the plasma membrane in APP/PS1 mice may be a contributing factor to the memory deficits with severe synapse loss that accompany this AD model.

### 3.1. Stable Expression Levels of mGlu_5_ Protein in the Hippocampus of APP/PS1 Mice

Pathological accumulation of Aβ plaques throughout the hippocampus in the APP/PS1 model of AD causes critical changes in the hippocampal circuits that include degradation of dendritic spines, reductions in synapse density, decreases in synaptic AMPA receptors and increases in the intrinsic excitability of neurons [34,35,36,37]. Therefore, glutamatergic neurons are critically affected in the hippocampus of APP/PS1 mice, and compiling evidence shows that mGlu_5_ has a contributing role in such neurodegenerative processes. A seminal report identified a molecular interaction between mGlu_5_ and cellular prion protein (PrP^C^), both acting as a co-receptor for oligomeric Aβ [26], and this macromolecular complex links mGlu_5_ to intracellular signalling molecules including Homer1b/c, Pyk2, Fyn, and CaMKII [26,38,39]. As these molecules play major roles in synaptic plasticity, the mGlu_5_-PrP^C^ complex mediates activation of Pyk2, Fyn and CaMKII following exposition to Aβ, which leads to impaired LTP [26,38,40].

The involvement of the glutamatergic system in AD is frequently paralleled with a regulation in the expression level of signalling molecules in given brain regions, as a consequence of their loss of function and neuronal cell death (reviewed by [41]). In this study, we detected no significant change in the total protein expression of mGlu_5_, or in its downstream molecules such as G_q/11_ and PLCβ_1_, in hippocampal plasma membranes of 12-month-old APP/PS1 mice. This is consistent with previous studies using similar methodological approaches and the same AD model, which showed no changes in mGlu_5_ expression levels at the same age [42,43]. Similarly, although translation of findings from animals into humans is relatively unsuccessful because animal models do not faithfully reproduce AD pathology, studies of postmortem human brain tissue also showed no changes in the expression level of mGlu_5_, G_q/11_ and PLCβ_1_ in the cerebral cortex in AD, although decreased binding to mGlu receptors and mGlu_1_ expression was reported [44]. On the other hand, expression of mGlu_5_ appears to decline across the lifespan, as shown by in situ hybridization in rats [45]. Consistent with this data, a recent study described a downregulation of mGlu_5_ in the brain of Senescence-accelerated mouse prone 8 strain (SAMP8), which represents a good model for accelerated senescence and to study the initial neurodegenerative alterations in AD [46]. The hippocampus of 5xFAD mice, which have large deposition of Aβ in the brain, has also showed reduced expression of mGlu_5_ compared to controls [47,48]. Therefore, the direct interaction between Aβ and mGlu_5_ to promote a downregulation of the receptor is dependent on the animal models of AD.

### 3.2. Postsynaptic Arrangement of mGlu_5_ in Principal Cells

This study provides detailed information on the organization of mGlu_5_ in the CA1 region and DG of the hippocampus in physiological and pathological conditions using immunogold particles of small size as a marker. A frequent limiting factor in immunoelectron microscopic studies is the sensitivity that often allows low immunoparticle counts [49]. In the present study, high sensitivity was achieved using the SDS-FRL technique, which offers a higher spatial resolution than standard embedding procedures, thus providing a two-dimensional distribution of receptors and ion channels and accurate data about their density along the neuronal surface [49,50].

The mGlu_5_ is the most abundant group I receptor subtype in the CA1 region of the hippocampus, as shown by previous in situ hybridization [19] and immunohistochemical studies [20,21,22,51], with a low expression for the mGlu_1_ subtype [52,53]. However, the DG shows high levels of both mGlu_1_ and mGlu_5_ transcript and protein [21,52]. Therefore, mGlu_5_ is thought to play a predominant role in regulating CA1 pyramidal cells, while mGlu_1_ and mGlu_5_ could contribute to the regulation of granule cells. Accordingly, our data show that mGlu_5_ was particularly strong in dendritic layers of the hippocampus in wild type and APP/PS1 mice.

At the ultrastructural level, surface-localised mGlu_5_ were found mainly on the extrasynaptic plasma membrane of dendritic spines and shafts and, at lower levels, on somata of pyramidal and granule cells. This agrees with previous reports showing that the highest density of mGlu_5_ was found in the perisynaptic region of dendritic spines of pyramidal cells [21], where they are generally recruited by the high levels of glutamate released during sustained synaptic transmission. Our data thus underlie the dominant postsynaptic role of mGlu_5_ as observed in previous electrophysiological studies [54]. The present quantification extends these findings and revealed that immunoparticles for mGlu_5_ were present on the somato-dendritic membrane of CA1 pyramidal cells and DG granule cells. Furthermore, mGlu_5_ immunoparticles were distributed in a non-uniform manner, with an increasing density from the soma to dendrites and to dendritic spines. Detailed electrophysiological investigations are needed to understand how the integration of signals in principal neurons of the hippocampus is affected by this non-uniform distribution of mGlu_5_. One possibility is that mGlu_5_ contributes to the activity of the hippocampal microcircuit by modifying synaptic plasticity [30,55].

A further interesting finding was the formation of clusters of mGlu_5_ immunoparticles along the neuronal surface. This preferential clustering distribution of mGlu_5_ on dendrites of principal cells offers an optimal position for the modulation of a variety of ion channels and neurotransmitter receptors residing in the dendritic compartments. For example, activation of mGlu_5_ in the hippocampus results in the modulation of small-conductance Ca^2+^-activated K+ (SK) channels [56], transient receptor potential C (TRPC) channels [57], L-type voltage-gated Ca^2+^ channels [58], as well as the ionotropic AMPA and NMDA receptors [59]. Thus, glutamate activation of mGlu_5_ can lead to multiple signalling mechanisms and signal transduction pathways, suggesting that any possible alteration in its subcellular localisation could alter glutamate signalling in the APP/PS1 model of AD in different ways.

### 3.3. Evidence for the Reduction of mGlu_5_ on the Surface of Principal Cells

The most significant finding of the present study is the reduction in the density of mGlu_5_ on the plasma membrane of CA1 pyramidal cells and DG granule cells in the hippocampus of APP/PS1 mice, accompanied by an increase in the frequency of immunoparticles at intracellular sites. These changes are not the result of neuronal loss associated with disease progression, because this AD animal model shows significant cell death only adjacent to plaques [60], and our study was carried out in Aβ plaque-free areas of the CA1 region in these animals. Recent studies reported similar plasma membrane to cytoplasm redistribution for GABA_B_ receptors and AMPA receptors in the same neuronal compartments [37,61,62].

Although no functional studies have been performed in the present study, several implications may be inferred from our data. Decreased surface expression of mGlu_5_ receptors may reduce cytoplasmic calcium (Ca^2+^) concentrations. In physiological conditions, mGlu_5_ stimulates release of Ca^2+^ from intracellular stores and such mobilization of intracellular Ca^2+^ contributes to regulate glutamatergic synaptic transmission in hippocampal neurons through the activation of SK and TRPC channels [57]. Interestingly, SK channels contributes to LTP in the CA1 region, where they interact with NMDA and mGlu_5_ in the same dendritic spines [56,63]. The induction of LTP requires NMDA receptor-mediated entry of Ca^2+^ into dendritic spines of pyramidal cells. Activation of mGlu receptors enhances NMDA receptor currents and facilitates the induction of LTP through PKC [64]. Our findings that postsynaptic mGlu_5_ are reduced in dendritic spines of CA1 pyramidal cells and DG granule cells, where they couple to NMDA receptors, suggest that mGlu_5_ might play a role in the alteration of LTP in these two hippocampal regions. This preferential role of mGlu_5_ is supported by results demonstrating that LTP can be induced in the CA1 region and the DG in mice lacking mGlu_1_, the other Group-I mGlu receptor [65].

In conclusion, we report here, for the first time, the ultrastructural distribution of mGlu_5_ along the somato-dendritic domains of pyramidal and granule cells in the hippocampus, and how this subcellular distribution is altered in APP/PS1 mice. Our findings highlight the functional importance of this redistribution as it implies a specific lack of functions for mGlu_5_ in the modulation of neuronal excitability in pathological conditions. This will provide a better understanding on the role of mGlu_5_ and associated proteins in pathological conditions and, ultimately, facilitate novel therapeutic approaches to target signalling molecules. Additional studies will further clarify the efficacy of these molecules in the progression of Alzheimer’s disease.

## 4. Material and Methods

### 4.1. Animals

Male APP/PS1 mice (RRID:IMSR_MMRRC:034832) were obtained from the Jackson Laboratory (https://www.jax.org/strain/005864) and expressed Mo/Hu APP695swe construct in conjunction with the exon-9-deleted variant of human presenilin 1 [Tg(APPswe,PSEN1dE9)85Dbo/Mmjax] [60,66]. The “control” wild type (WT) mice were age-matched littermates without the transgene. The following ages were selected for analysis: (i) no sign of pathology (1 month), (ii) first signs of Aβ deposition (6 months) [60] and (iii) onset of memory deficits with severe synapse loss and widespread Aβ deposition (12 months). For all ages and genotypes, mice were used as follows for the experiments: Immunoblot (4), Histoblot (4), SDS-digested freeze-fracture replica labelling (SDS-FRL) (4) and pre-embedding immunogold experiments (3). All mice were maintained at the Animal House Facility of the University of Castilla-La Mancha (Albacete, Spain) in cages of 2 or more mice, on a 12-h light/12-h dark cycle at 24 °C and received food and water ad libitum. Care and handling of animals prior to and during experimental procedures were in accordance with Spanish (RD 1201/2015) and European Union regulations (86/609/EC), and all protocols and methodologies were approved by the local Animal Care and Use Committee.

For immunoblotting and histoblotting, animals were deeply anesthetised by intraperitoneal injection of ketamine/xylazine 1:1 (ketamine, 100 mg/kg; xylazine, 10 mg/kg). The hippocampus was dissected, frozen rapidly in liquid nitrogen and stored at −80 °C. For immunohistochemistry experiments at both the light microscopic and electron microscopic level, using the pre-embedding immunogold technique, animals were firstly deeply anaesthetised by intraperitoneal injection of ketamine/xylazine 1:1 (ketamine, 100 mg/kg; xylazine, 10 mg/kg) and then transcardially perfused with ice-cold fixative containing 4% (*w*/*v*) paraformaldehyde with 0.05% (*v*/*v*) glutaraldehyde in 0.1 M phosphate buffer (PB, pH 7.4) for 15 min. After perfusion, brains were removed and immersed in the same fixative for 2 h or overnight at 4 °C. Tissue blocks were washed thoroughly in 0.1 M PB. Coronal sections (60 µm thickness) were cut using a Vibratome (Leica V1000, Leica, Wetzlar, Germany). For SDS-FRL, see below.

### 4.2. Antibodies and Chemicals

For immunoblots, we used rabbit anti-mGlu_5_ polyclonal antibody (1:500, GTX133288, Genetex, Inc., Alton Pkwy Irivine, CA, USA), rabbit anti-G_q/11_ polyclonal antibody (1:500, 06-709, Upstate Biotechnology Inc., Lake Placid, NY, USA), mouse anti-PLCβ_1_ monoclonal antibody (1:500, 05-164, Upstate), and mouse anti-β tubulin monoclonal antibody (1:2000, 05-661, Upstate). For SDS-FRL and pre-embedding immunogold labelling, we used rabbit anti-mGlu_5_ polyclonal antibody (Rb-Af300; aa. 1144–1171 of mouse mGlu_5_; RRID: AB_2571802; Frontier Institute Co., Sapporo, Japan) and a guinea pig anti-mGlu_5_ polyclonal antibody (GP-Af270; aa. 1144–1171 of mouse mGlu_5_; RRID: AB_2571804; Frontier Institute Co., Sapporo, Japan). The preparation, purification and full characterization of these antibodies has been described previously [67,68]. The specificity of the rabbit anti-mGlu_5_ polyclonal antibody (Rb-Af300) using SDS-FRL in mGluR5 KO mice has been described previously [56].

The secondary antibodies used were as follows: goat anti-mouse IgG-horseradish peroxidase (1:2000; Santa Cruz Biotechnology, Santa Cruz, CA, USA), goat anti-rabbit IgG-horseradish peroxidase (1:15,000; Thermo Fisher Scientific, Waltham, MS, USA), alkaline phosphatase (AP)-goat anti-mouse IgG (H+L) and AP-goat anti-rabbit IgG (H+L) (1:5000; Invitrogen, Paisley, UK), anti-rabbit IgG conjugated to 10 nm gold particles and anti-guinea IgG conjugated to 10 nm gold particles (1:100; British Biocell International, Cardiff, UK), anti-rabbit IgG coupled to 1.4 nm gold or anti-guinea IgG coupled to 1.4 nm gold (Nanoprobes Inc., Stony Brook, NY, USA).

### 4.3. Immunoblots

Hippocampi were homogenised in Tris buffer (TB; 50 mM Tris-HCl, pH 7.4), containing protease inhibitor cocktail (Thermo Fisher Scientific, Waltham, MS, USA), with a motorised pestle (Sigma-Aldrich, St. Louis, MO, USA). The homogenised tissue was initially centrifuged 10 min at 1000× *g* at 4 °C and the supernatant was further centrifuged 30 min at 12,000× *g* (Centrifuge 5415R, Eppendorf, Hamburg, Germany) at 4 °C. The resulting pellet, containing the membrane extracts, was resuspended in the TB. The protein content of each membrane extract was determined by the BCA protein assay kit (Thermo Fisher Scientific, Waltham, MS, USA). Forty micrograms of membrane protein were loaded onto sodium dodecyl sulphate polyacrylamide (7.5%) gels (SDS/PAGE) in sample buffer (0.05 M Tris pH 6.8, 2% (*w*/*v*) SDS, 10% (*v*/*v*) glycerol, 0.05% (*v*/*v*) *β*-mercaptoethanol and 0.001% (*w*/*v*) bromophenol blue). The proteins were transferred to Nitrocellulose membranes using the iBlot^TM^ Dry Blotting System (Invitrogen, Madrid, Spain), followed by immunolabelling with anti-mGlu_5_ (1:500), anti-G_q/11_ (1:500), anti-PLCβ_1_ (1:500) and anti-β-tubulin (1:2000) antibodies. Protein bands were visualized after application of a rabbit IgG kappa binding protein coupled to horseradish peroxidase (1:2000) using the enhanced chemiluminescence (ECL) blotting detection kit (GE Healthcare, Madrid, Spain) in a G:Box gel documentation system (Syngene, Cambridge, UK), and specific bands were quantified by densitometry using GeneTools software (Syngene). A series of primary and secondary antibody dilutions and incubation times were used to optimize the experimental conditions for the linear sensitivity range, confirming that our labelling was well below saturation levels.

### 4.4. Histoblotting

The regional distribution of mGlu_5_ was analysed in rodent brains, using the histoblot technique [69]. Briefly, horizontal cryostat sections (10 µm) from mouse brain were overlapped with nitrocellulose membranes moistened with 48 mM Tris-base, 39 mM glycine, 2% (*w*/*v*) sodium dodecyl sulphate and 20% (*v*/*v*) methanol for 15 min at room temperature (~20 °C). After blocking in 5% (*w*/*v*) non-fat dry milk in phosphate-buffered saline for 1 h, nitrocellulose membranes were treated with DNase I (5 U/mL), washed and incubated in 2% (*w*/*v*) sodium dodecyl sulphate and 100 mM β-mercaptoethanol in 100 mM Tris–HCl (pH 7.0) for 60 min at 45 °C to remove adhering tissue residues. After extensive washing, the blots were reacted with affinity-purified anti-mGlu_5_ antibodies (0.5 mg/mL) in blocking solution overnight at 4 °C. The bound primary antibodies were detected with alkaline phosphatase-conjugated anti-rabbit IgG secondary antibodies [69]. A series of primary and secondary antibody dilutions and incubation times were used to optimize the experimental conditions for the linear sensitivity range of the alkaline phosphatase reactions. To compare the expression levels of the mGlu_5_ receptor between the two genotypes (wild type and APP/PS1) and ages (1, 6 and 12-months), all nitrocellulose membranes were processed in parallel, and the same incubation time for each reagent was used for the antibody. Digital images were acquired by scanning the nitrocellulose membranes using a desktop scanner (HP Scanjet 8300). Image analysis and processing were performed using the Adobe Photoshop software (Adobe Systems, San Jose, CA, USA) as described previously [62].

### 4.5. Immunohistochemistry for Electron Microscopy

Immunohistochemical reactions at the electron microscopic level were carried out using the pre-embedding immunogold and the SDS-FRL methods, as described earlier [21,27].

*Pre-embedding immunogold method*. Briefly, free-floating sections obtained from the two genotypes (WT and APP/PS1) and three ages (1, 6 and 12-months) were incubated in parallel in 10% (*v*/*v*) NGS diluted in Tris buffered saline (TBS; 50 mM Tris-HCl, 0.9% NaCl, pH 7.4). Sections were then incubated in anti-mGlu_5_ antibodies (3–5 μg⁄ mL diluted in TBS containing 1% (*v*/*v*) NGS), followed by incubation in goat anti-rabbit IgG or goat anti-guinea pig IgG coupled to 1.4 nm gold (Nanoprobes Inc., Stony Brook, NY, USA). Sections were post-fixed in 1% (*v*/*v*) glutaraldehyde and washed in double-distilled water, followed by silver enhancement of the gold particles with an HQ Silver kit (Nanoprobes Inc.). Sections were then treated with osmium tetraoxide (1% in 0.1 M phosphate buffer), block-stained with uranyl acetate, dehydrated in graded series of ethanol and flat-embedded on glass slides in Durcupan (Sigma-Aldrich, St. Louis, MO, USA) resin. Regions of interest were cut at 70–90 nm on an ultramicrotome (Reichert Ultracut E, Leica, Vienna, Austria) and collected on single slot pioloform-coated copper grids. Ultrastructural analyses were performed in a JEOL-1010 electron microscope (JEOL Ltd., Tokyo, Japan).

*SDS-FRL technique*. Animals were anesthetised with sodium pentobarbital (50 mg/kg, i.p.) and perfused transcardially with saline for 1 min, followed by perfusion with 2% paraformaldehyde in 0.1 M phosphate buffer (PB) for 12 min. The hippocampi were dissected and cut into sagittal slices (130 µm) using a microslicer (Dosaka, Kyoto, Japan) in 0.1 M PB. Hippocampal sections containing the CA region were then immersed in graded glycerol of 10–30% in 0.1 M PB at 4 °C overnight. Slices were frozen using a high-pressure freezing machine (HPM010, BAL-TEC, Balzers). Slices were then fractured into two parts at −120 °C and replicated by carbon deposition (5 nm thick), platinum (60° unidirectional from horizontal level, 2 nm), and carbon (15 nm) in a freeze-fracture replica machine (BAF060, BAL-TEC, Balzers). Replicas were transferred to 2.5% SDS and 20% sucrose in 15 mM Tris buffer (pH 8.3) for 18 h at 80 °C with shaking to dissolve tissue debris. The replicas were washed three times in TBS containing 0.05% bovine serum albumin (BSA), and then blocked with 5% BSA in the washing buffer for 1 h at room temperature. Next, the replicas were washed and reacted with a polyclonal rabbit antibody for mGlu_5_ (5 μg/mL) at 15 °C overnight. Following three washes in 0.05% BSA in TBS and blocking in 5% BSA/TBS, replicas were incubated in secondary antibodies conjugated with 10 nm gold particles overnight at room temperature. After immunogold labelling, the replicas were immediately rinsed three times with 0.05% BSA in TBS, washed twice with distilled water, and picked up onto grids coated with pioloform (Agar Scientific, Stansted, Essex, UK).

### 4.6. Quantification and Analysis of SDS-FRL Data

The labelled replicas were examined using a transmission electron microscope (JEOL-1010) and images captured at magnifications of 80,000, and 100,000. The antibody used in this study was visualised by immunoparticles on the protoplasmic face (P-face), consistent with the intracellular location of the epitope. Non-specific background labelling for mGlu_5_ was estimated by counting immunogold particles on the exoplasmic face (E-face) surfaces in wild type mice. Digitized images were then modified for brightness and contrast using Adobe PhotoShop CS5 (Mountain View, CA, USA) to optimize them for quantitative analysis.

*Density gradient of mGlu_5_ along the neuronal surface.* The procedure was similar to that which was used previously [62]. Briefly, immunogold labelling for mGlu_5_ was achieved from replicas containing all layers of the CA1 region and DG, so that the laminar distribution could be compared under identical conditions for each animal and experimental group. Quantitative analysis of immunogold labelling for mGlu_5_ was performed on 7 different dendritic compartments of pyramidal cells in all dendritic layers of the CA1 region, the somata of pyramidal cells in the *stratum pyramidale*, as well as on 2 dendritic compartments of granule cells in the molecular layer and granule cell layer of the DG. The dendritic compartments analysed in principal cells were the main dendritic shaft (apical dendrites), spiny branchlets (oblique dendrites) and dendritic spines. Oblique dendrites were identified based on their small diameter and the presence of at least one emerging spine from the dendritic shaft. Dendritic spines were considered as such if: (i) they emerged from a dendritic shaft, or (ii) they opposed an axon terminal. Axon terminals were identified based on: (i) the presence of an active zone (AZ) facing a postsynaptic density (PSD) recognised by an accumulation of intramembrane particles (IMPs) on the opposing exoplasmic-face (E-face) of a spine or dendrite; or (ii) the presence of synaptic vesicles on their cross-fractured portions. Non-specific background labelling was measured on E-face structures surrounding the measured P-faces. Images of the identified compartments were selected randomly over the entire dendritic tree of CA1 pyramidal cells and DG granule cells, and then captured with an ORIUS SC1000 CCD camera (Gatan, Munich, Germany). The area of the selected profiles and the number of immunoparticles were measured using our GPDQ software [27]. Immunoparticle densities were presented as mean ± SEM between animals. Statistical comparisons were performed with GraphPad Prism 5 software (La Jolla, CA, USA).

Given that neuronal loss is observed adjacent to plaques in APP/PS1 mice [35], our quantitative analysis was performed in Aβ plaque-free regions of the hippocampus to avoid the destroyed tissue in dystrophic neurites adjacent to Aβ plaques. Thus, the density values expressed as immunoparticles/µm^2^ in the APP/PS1 mice represent genuine reductions in mGlu_5_ receptors in different compartments of CA1 pyramidal cells and DG granule cells, regardless of any possible neuronal and/or synaptic loss.

### 4.7. Controls

To test method specificity in the procedures for electron microscopy, the primary antibody was either omitted or replaced with 5% (*v*/*v*) normal serum of the species of the primary antibody, resulting in total loss of the signal. For the pre-embedding technique, labelling patterns were also compared with those obtained by calbindin (polyclonal rabbit anti-calbindin D-9k CB9; Swant, Marly, Switzerland); only the antibodies against mGlu_5_ consistently labelled excitatory synapses.

### 4.8. Data Analysis

To avoid observer bias, we performed blinded experiments for immunoblots and immohistochemistry prior to data analysis. Statistical analyses were performed using GraphPad Prism (San Diego, CA, USA) and data were presented as mean ± SEM unless indicated otherwise. Statistical significance was defined as *p* < 0.05. The statistical evaluation of the immunoblots was performed using the Student–Fisher *t*-test, with Levene’s test for homogeneity of variance and the Shapiro–Wilk normality test. The application of Levene’s test indicated that variances are equal, and the application of the Shapiro–Wilk test indicated that distributions were normal. To compute SEM error bars, five blots were measured from each animal. The statistical evaluation of the immunogold densities in the mouse model was performed using the two-way ANOVA test and Bonferroni post hoc test. Finally, statistical evaluation of the frequency of immunogold measured with pre-embedding were performed using the Student-*t* test with Holm–Sidak correction.

## Figures and Tables

**Figure 1 ijms-22-05867-f001:**
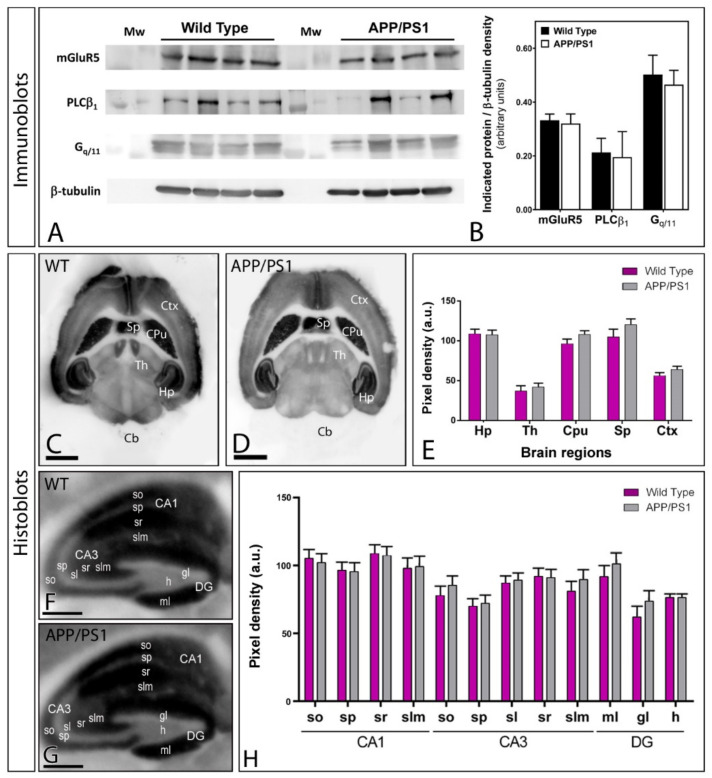
Expression of mGlu_5_ in wild type and APP/PS1 mice. (**A**,**B**) Immunoblots showing the expression of mGlu_5_, PLCβ_1_ and G_q/11_ in hippocampal membranes from APP/PS1 and wild type mice at 12 months of age. The developed immunoblots were scanned and densitometric measurements from three independent experiments were averaged together. Quantification of the three proteins were normalised to β-tubulin and expressed as pixel density, showing no differences in APP/PS1 mice compared to age-matched wild type controls. (**C**–**E**) The regional brain mGlu_5_ expression was visualised in histoblots of horizontal brain sections at 12 months of age in wild type and APP/PS1 mice using an affinity-purified anti-mGlu_5_ antibody. The expression of mGlu_5_ in different brain regions was determined by densitometric analysis of the scanned histoblots. The expression of mGlu_5_ revealed marked region-specific differences, with the strongest immunoreactivity in the hippocampus (Hp), caudate putamen (CPu) and septum (Sp), and moderate labelling in the cortex (Ctx) and thalamus (Th). The weakest expression level was detected in the cerebellum (Cb). Densitometric analysis showed no differences in mGlu_5_ expression in APP/PS1 mice compared to age-matched wild type controls at the ages employed. (**F**,**G**) Hippocampal expression and distribution of mGlu_5_ in wild type and APP/PS1 mice visualised in histoblots of horizontal sections at 12 months of age. (**H**) Expression for mGlu_5_ was strong in all dendritic layers of the CA1 and CA3 region and DG, with the *strata oriens* (so) and *radiatum* (sr) of the CA1 region showing the highest expression levels. A more moderate expression was observed in the *stratum lacunosum-moleculare* (slm) of CA1, and the *strata oriens* (so), *radiatum* (sr) and *lacunosum-moleculare* (slm) of CA3, with the molecular layer (ml) and hilus (h) of the DG showing the lowest expression level. Densitometric analysis showed no differences in mGlu_5_ expression in APP/PS1 mice compared to age-matched wild type controls. Error bars indicate SEM. Scale bars: (**C**,**D**), 0.2 cm; (**F**,**G**), 0.05 cm.

**Figure 2 ijms-22-05867-f002:**
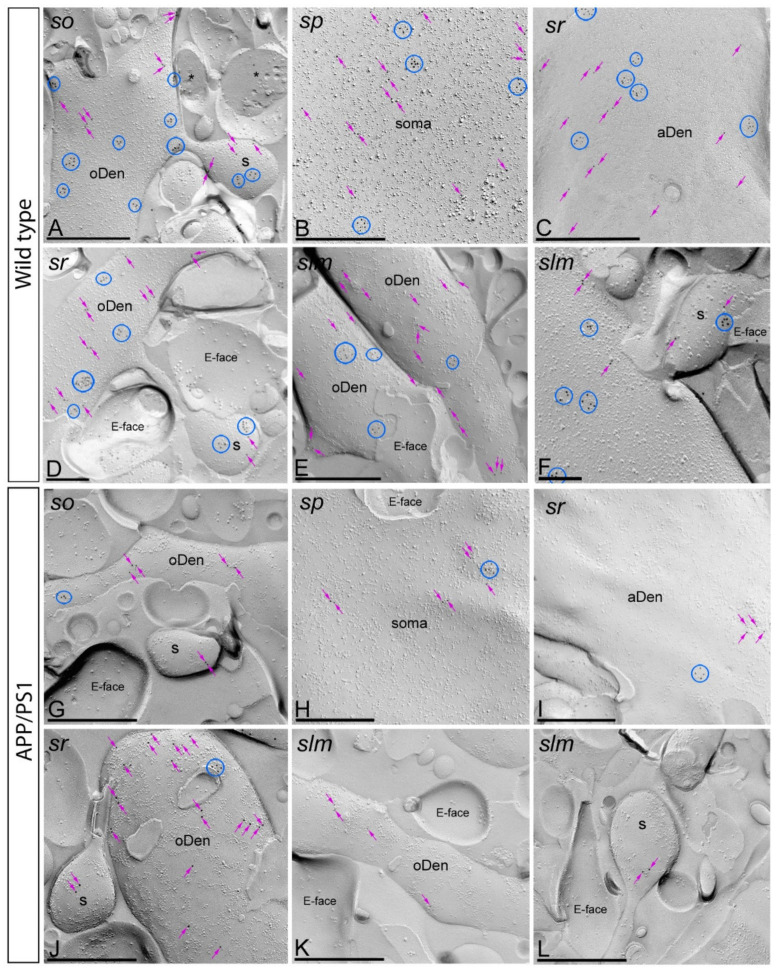
Subcellular localisation of mGlu_5_ in somato-dendritic domains of CA1 pyramidal cells in wild type and APP/PS1 mice. (**A**–**F**) Electron micrographs obtained from different subfields of the CA1 region showing immunoparticles for mGlu_5_ along the surface of pyramidal cells, as detected using the SDS-FRL technique in wild type mice at 12 months of age. Immunoparticles for mGlu_5_ were detected forming clusters (blue circles) or scattered (purple arrows) associated with the P-face in the soma, apical dendrites (aDen), oblique dendrites (oDen) and dendritic spines (s) of CA1 pyramidal cells. Cross-fractures are indicated with asterisks (*). The E-face is free of mGlu_5_ immunolabelling. (**G**–**L**) Electron micrographs obtained from different subfields of the CA1 region showing immunoparticles for mGlu_5_ along the surface of pyramidal cells, as detected using the SDS-FRL technique in APP/PS1 mice at 12 months of age. Immunoparticles for mGlu_5_ were detected, at low frequency, forming clusters (blue circles) or scattered (purple arrows) associated with the P-face in the soma, apical dendrites (aDen), oblique dendrites (oDen) and dendritic spines (s) of CA1 pyramidal cells. Cross-fractures are indicated with asterisks (*). The E-face is free of mGlu_5_ immunolabelling. Abbreviations: *so*, *stratum oriens*; *sp*, *stratum pyramidale*; *sr*, *stratum radiatum*; *slm*, *stratum lacunosum-moleculare*. Scale bars: (**A**–**C**,**E**,**G**–**L**) 0.5 μm; (**D**,**F**), 0.2 μm.

**Figure 3 ijms-22-05867-f003:**
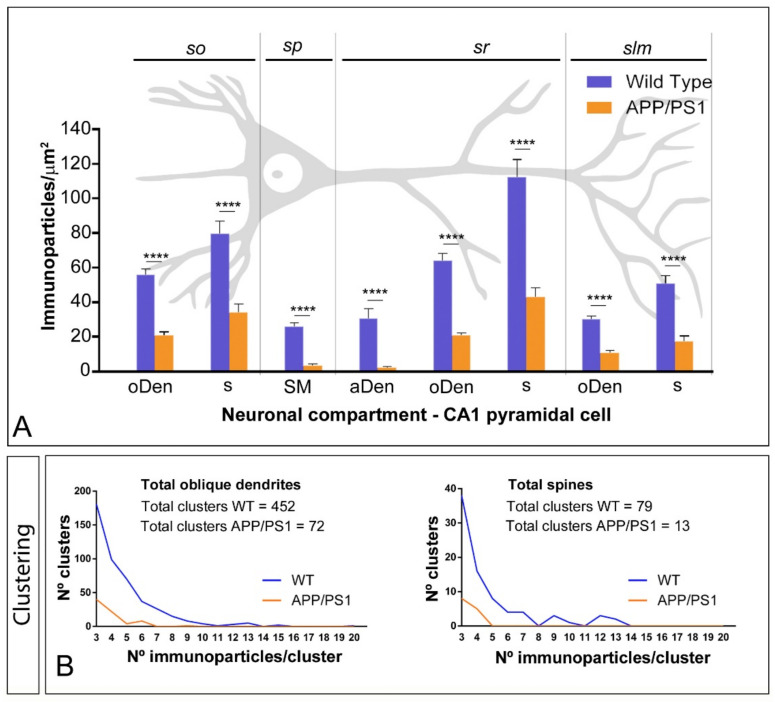
Quantitative analysis of mGlu_5_ immunoparticles along the surface of CA1 pyramidal *cells*. (**A**) Density gradient of mGlu_5_ immunogold labelling in eight neuronal compartments of CA1 pyramidal cells in wild type and APP/PS1 mice at 12 months. The density gradient of surface mGlu_5_ immunoparticles was significantly reduced in the APP/PS1 mice compared to age-matched wild type controls in all strata and subcellular compartments analysed (Two-way ANOVA test and Bonferroni post hoc test, **** *p* < 0.0001) (Table 1). Error bars indicate SEM. Abbreviations: *so*, *stratum oriens*; *sp*, *stratum pyramidale*; *sr*, *stratum radiatum*; *slm*, *stratum lacunosum-moleculare*; aDen, apical dendrite; oDen, oblique dendrite; s, spine; SM, soma. (**B**) Histograms showing composition of clusters in all oblique dendrites and all spines of CA1 pyramidal cells. For all of the oblique dendrites analysed, the number and size of clusters of mGlu_5_ immunoparticles was reduced in the APP/PS1 mice (72 clusters with a range of 3 to 9 immunoparticles in oblique dendrites; 13 clusters with a range of 3 to 4 immunoparticles in spines) compared to age-matched wild type controls (452 clusters with a range of 3 to 20 immunoparticles in oblique dendrites; 79 clusters with a range of 3 to 13 immunoparticles in spines).

**Figure 4 ijms-22-05867-f004:**
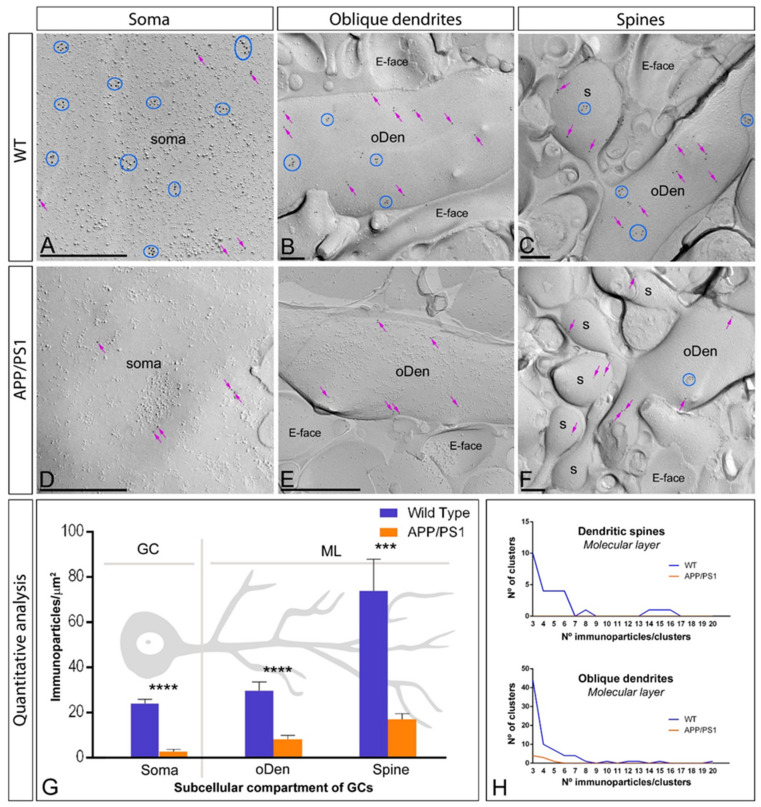
Subcellular localisation of mGlu_5_ in somato-dendritic domains of DG granule cells in wild type and APP/PS1 mice. (**A**–**F**) Electron micrographs of the DG showing immunoparticles for mGlu_5_ along the surface of granule cells, as detected using the SDS-FRL technique in wild type (panels **A**–**C**) and APP/PS1 mice (panels **D**–**F**) at 12 months of age. Immunoparticles for mGlu_5_ were detected forming clusters (blue circles) or scattered (purple arrows) associated with the P-face in the soma, oblique dendrites (oDen) and dendritic spines (s) of granule cells cells. (**G**) Quantitative analysis showing that the density gradient of surface mGlu_5_ immunoparticles was significantly reduced in the APP/PS1 mice compared to age-matched wild type controls in all strata and subcellular compartments analysed (Two-way ANOVA test and Bonferroni post hoc test, *** *p* < 0.001, **** *p* < 0.0001) (Table 1). Error bars indicate SEM. (**H**) Histograms showing composition of clusters in dendritic spines and oblique dendrites of granule cells. The composition of clusters was reduced in the APP/PS1 mice (1 cluster with 3 immunoparticles in spines; 8 clusters with a range of 3 to 5 immunoparticles in oblique dendrites) compared to age-matched wild type controls (26 clusters with a range of 3 to 16 immunoparticles in spines; 75 clusters with a range of 3 to 20 immunoparticles in oblique dendrites). Scale bars: (**A**,**D**), 0.5 μm; (**B**,**C**,**E**,**F**), 0.2 μm.

**Figure 5 ijms-22-05867-f005:**
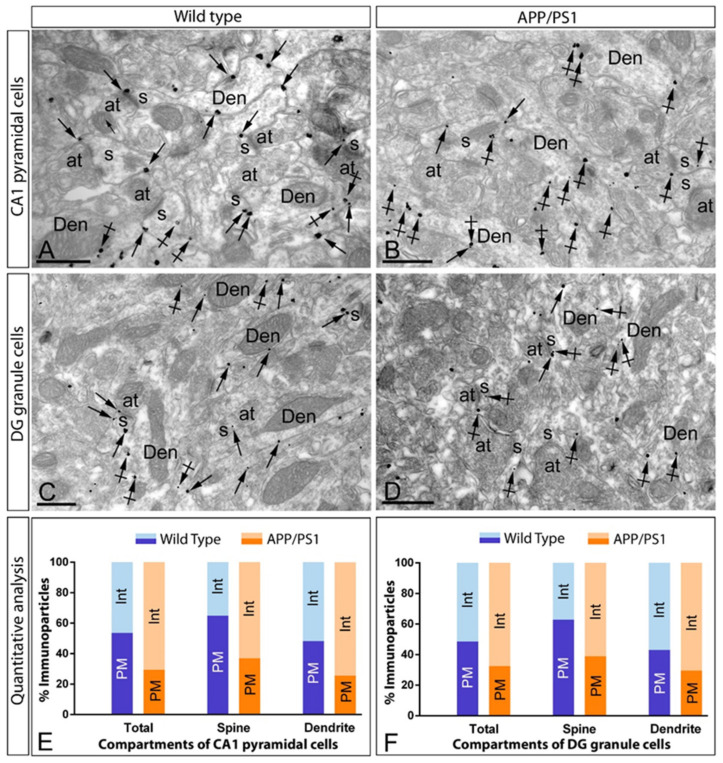
Intracellular distribution of mGlu_5_ is increased in principal cells of APP/PS1 mice. Electron micrographs showing immunoparticles for mGlu_5_ in the *stratum radiatum* of the CA1 region and molecular layer of the DG at 12 months of age in wild type and APP/PS1 mice, as detected using a pre-embedding immunogold technique. (**A**–**D**) In wild type mice, mGlu_5_ immunoparticles were mostly located at the extrasynaptic membrane (arrows) of dendritic shafts (Den) and dendritic spines (s) of pyramidal and granule cells, and less frequently at intracellular sites (arrowheads). In APP/PS1 mice, however, fewer mGlu_5_ immunoparticles were observed along the extrasynaptic membrane (arrows) of dendritic shafts (Den) and dendritic spines (s), while more immunoparticles were observed at intracellular sites (arrowheads). (**E**,**F**) Quantitative analysis at 12 months of age demonstrating that mGlu_5_ immunoparticles were less frequently observed along the extrasynaptic plasma membrane of dendrites and spines of CA1 pyramidal cells and DG granule cells, and more frequently at intracellular sites in APP/PS1 mice. Scale bars: (**A**–**D**), 1 μm.

**Table 1 ijms-22-05867-t001:** Distribution of mGlu_5_ immunoparticles along the membrane surface of CA1 pyramidal cells and DG granule cells at 12 months of age.

		Wild Type	APP/PS1
		Mean ± SEM (Immunoparticles/µm^2^)	Mean ± SEM (Immunoparticles/µm^2^)
CA1	*Stratum oriens*		
	Oblique Dendrites	56.02 ± 3.16	20.83 ± 2.01
	Spines	79.54 ± 7.37	34.05 ± 4.93
	*Stratum pyramidale*		
	Soma	26.23 ± 1.75	3.57 ± 0.92
	*Stratum radiatum*		
	Apical Dendrites	30.58 ± 5.54	2.43 ± 0.47
	Oblique Dendrites	64.26 ± 3.91	20.79 ± 1.68
	Spines	112.18 ± 10.26	43.23 ± 5.00
	*Stratum lacunosum-moleculare*		
	Oblique Dendrites	30.32 ± 2.00	11.25 ± 1.24
	Spines	50.90 ± 4.51	17.27 ± 3.15
DG	*Granule cell layer*		
	Soma	24.01 ± 1.84	2.70 ± 0.98
	*Molecular layer*		
	Oblique Dendrites	29.79 ± 3.75	8.25 ± 1.59
	Spines	73.86 ± 14.00	17.07 ± 2.43

**Table 2 ijms-22-05867-t002:** Composition of clusters of mGlu_5_ immunoparticles along the membrane surface of CA1 pyramidal cells and DG granule cells at 12 months of age.

Wild Type				APP/PS1			
No. Profiles	Area Profiles	No. Clusters	Range Gold Particles	No. Profiles	Area Profiles	No. Clusters	Range Gold Particles
24	22 µm^2^	184	13–3	24	21 µm^2^	27	6–3
24	1.8 µm^2^	29	13–3	24	1.7 µm^2^	6	4–3

15	17 µm^2^	74	11–3	15	21 µm^2^	8	10–3

15	17 µm^2^	38	11–3	15	21 µm^2^	9	5–3
24	20 µm^2^	209	20–3	24	18 µm^2^	34	9–3
30	2 µm^2^	33	9–3	30	1.9 µm^2^	7	4–3

24	15 µm^2^	59	9–3	24	14 µm^2^	14	5–3
27	2 µm^2^	17	5–3	27	1.9 µm^2^	0	
183	97 µm^2^	643		183	100 µm^2^	103	

10	14 µm^2^	49	13–3	10	13 µm^2^	2	3–3

18	22 µm^2^	75	20–3	18	21 µm^2^	8	5–3
21	4 µm^2^	26	16–3	21	4 µm^2^	0	0
49	40 µm^2^	150		49	38 µm^2^	11	

## Data Availability

All data used and/or analysed during the current study are available from the corresponding author on reasonable request.

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
