# Peer review of "The Density of Group I mGlu5 Receptors Is Reduced along the Neuronal Surface of Hippocampal Cells in a Mouse Model of Alzheimer’s Disease"

_ijms, 2021, doi:10.3390/ijms22115867_

Round 1
Reviewer 1 Report
Please see an attached file.

Author Response
We are grateful for the valuable suggestions of the Referee that we have incorporated into the manuscript. The changed words have been highlighted in bold in the manuscript.
Q 1 - Specific comment: “The usage of the word “mGlu5” is not consistent through the manuscript. “metabotropic glutamate receptor subtype 5” is abbreviated to “mGlu5” in Line 19, but in the main text, “mGlu5 receptor” is often used and it seems like a duplication of “receptor”. I suggest using “mGlu5” as the receptor and use “mGlu5 protein” when it is needed in the entire manuscript as authors did in the abstract”.
Authors’ response: The current nomenclature approved by IUPHAR is “metabotropic glutamate receptor subtype 5” and “mGlu5 receptor” when written as abbreviations. However, the Reviewer is right when asking for more consistent usage of terminology. Therefore, following the suggestion of the Reviewer, we have changed terminology and used “mGlu5”, and less frequently “mGlu5 protein”, throughout the text, with the only exception of the title.
Q 2 - Specific comment: “The immunoblot was performed from hippocampal membrane fraction. If so, I think the amount of mGlu5 in APP/PS1 should be less than control. Or, are the intracellular proteins also bound to membrane? From the preembedding EM images (Fig. 6), it doesn’t seem to be the case, and authors also state “immunoparticles at cytoplasmic sites” (line 392). Some discussions are appreciated”.
Authors’ response: We acknowledge the Reviewer for the observation. The immunoblots were performed from hippocampal membrane fraction. Immunoparticles detected at intracellular sites were mostly associated with intracellular membranes, likely small cisterns of the ER. This may not be clearly visible for every single gold particle, due to not so well preservation of ultrastructure. The way that normally the laboratories proceed when comparing expression levels of signalling molecules between WT and APP/PS1 or other models of AD is using the same amount of protein for each experimental group. We have used “cytoplasmic” as a similar terminology than “intracellular”. We have now replaced “cytoplasmic” for “intracellular” throughout the text. In addition, we have changed the original sentence to “…at intracellular sites associated with intracellular membranes…” to clarify the meaning of intracellular labelling.
Q 3 - Specific comment: “Related to the (3), in the abstract, authors state “mGlu5 “from” the plasma membrane to intracellular site” (lines 32-33). I am not sure why it can be said “from” membrane. Is there any possibilities that newly synthesized mGlu5 is not recruited to the plasma membrane? Additional discussion is appreciated”.
Authors’ response: We cannot rule out the possibility raised by the Reviewer. To avoid speculations in the text we have re-written the sentence to “… a loss of mGlu5 at the plasma membrane and accumulation at intracellular sites…”.
Q 4 - Specific comment: “In the SDS-FRL images (Figs. 2, 3, 5), I thought authors marked all of the immunoparticles with either arrows or circles However, I found quite a lot of immunoparticles unmarked. I tried to mark all of them (please see below), and the main finding of the reduction of mGlu5 in APP/PS1 seemed to be more significant. I suggest adding all marks, and if they are missed from the numerical analysis, it should be re-analyzed”.
Authors’ response: We acknowledge the Reviewer for the observation. Our intention was not marking all immunoparticles to avoid the use of too many arrows. Therefore, this was done on purpose trying to mark only some examples of immunoparticles. However, following the suggestion of the Reviewer, we have now marked all immunoparticles. Finally, to clarify, all immunoparticles present in sample images and detected by our software (and confirmed visually by the observer) were included in all quantitative analyses. Therefore, there is no need to re-analysed data.
Q 5 - Specific comment: “In Fig. 2F, the circle only contains 2 immunoparticles and should not be categorized as a cluster. Please delete the circle, or replace with another image having clustered particles”.
Authors’ response: We acknowledge the Reviewer for the observation. We have replaced the image.
Q 6 - Specific comment: “Lines 172-173; it says “immunoparticles for mGlu5 were mostly found forming clusters. Less frequently mGlu5 immunpparticles were also distributed as a scattered pattern”. I suggest including the total number of analyzed immunoparticles in total P-face, and/or the percentage of immunoparticles in clusters vs. scattered would make the finding clearer (as authors stated the particle percentage for the preembedding EM data in lines 277-282)”.
Authors’ response: We acknowledge the Reviewer for this interesting observation. Following this suggestion we have added the number of immunoparticles distributed following clustered and scattered pattern, in page 7 and 8.
Q 7 - Specific comment: “Line 178, “No immunolabeling was detected… axon terminals”. I don’t see an example of axon terminal (P-face) lacking immunoparticles in Fig. 2A-F. Please add an image if this is claimed”.
Authors’ response: We acknowledge the Reviewer for the observation. As the Reviewer must be aware, it is not particularly easy to have the P-face of both dendritic spines and associated axon terminals. The observation about axon terminals free of mGlu5 immunolabelling is based in several putative axon terminals, many of which were identified using double labelling with SNAP-25, a marker of axon terminals, and mGluR5. In such experiments we could see only labelling for SNPA-25. However, this set of experiments still need to the replicated, hence we did not want to include such data in the manuscript and the reason why we still would prefer not to do it. Therefore, to meet the requirement of the Reviewer, we have deleted “axon terminals” from the original sentence.
Q 7 - Specific comment: Minor points:
- In a subtitle, line 148, the letter “l” in “mGlu5” is missed.
Response: Changed.
- I suggest adding “in APP/PS1 mice” in the subtitles 2.3 (line 148) and 2.4 (line
229).
Response: Done.
- Regarding to the statement about “E-face having no signal”, I would like to point out one thing; some E-faces (usually small) when they are on a P-face that are labeled, like a small E-face patch at the lower center in Fig. 2E marked as “Eface”, could have cryptic labeling because the epitope of the antibody is beneath the P-face below the E-face. Actually, one immunoparticle at the edge of another E-face on the upper profile would be a signal (marked in below). As it is a known fact, I recommend stating particularly for this case, “E-faces of pyramidal cells are free of mGlu5 immunoparticles”.
Response: We acknowledge the Reviewer for this interesting observation. We fully agree on this observation. We have changed the sentence to that suggested by the Reviewer.
- I found the secondary antibody used in the experiments contained very small sizes of particles, which made the judgment of signal vs. noise vs. replica’s shadow much harder. I had some that I could not decide if they should be marked or not in the suggested figures below. It is caused by the production batch of the secondary antibody, and I recommend adding one sentence for the size variability of the immunoparticles in the method to avoid confusions.
Response: We acknowledge the Reviewer for this interesting observation. We have added a sentence at the end of the paragraph describing secondary antibodies in page 16-17.
Reviewer 2 Report
The research article by Rafael Luján and coworkers investigates expression profile the Metabotropic glutamate receptor subtype 5 and correlate with Alzheimer´s disease. The article is very interesting. However, before it could be considered for publication, authors need to incorporate some changes and revise their manuscript accordingly.
comments-
Abstract Highlight essentialities and future perspectives of the study.
Section Introduction - The authors are advised to cut short the introduction section of the manuscript.
Section results
Fig 1: Fig 1a and 1b: Is the difference statistically significant as very little difference has been observed between wild type and APP/PS1. Fig 1H: The bars are too small compared to 1b and 1E. Same pattern need to be followed.
Fig 2 & 3: It is better to make their appearance as comparative one so that the difference is visualized from their comparison.
Fig 4: Information pertaining to Fig 4 should follow the combination of fig 2 & 3 in a similar manner as fig 5.
Discussion:
The sections need to be revised as per the revised figures.
Section conclusion: A short section highlighting importance of the study and future directions with possible limitations would add strength to the study undertaken.
Author Response
We are grateful for the valuable suggestions of the Referee that we have incorporated into the manuscript. The changed words have been highlighted in bold in the manuscript.
Q 1 - Specific comment: “Abstract - Highlight essentialities and future perspectives of the study”.
Authors’ response: Following the suggestion of the reviewer, we have added a couple of sentences with some of the information requested.
Q 2 - Specific comment: “Section Introduction - The authors are advised to cut short the introduction section of the manuscript”.
Authors’ response: Following the suggestion of the reviewer, we have shortened the introduction, deleting several sentences.
Q 3 - Specific comment: “Fig 1: Fig 1a and 1b: Is the difference statistically significant as very little difference has been observed between wild type and APP/PS1. Fig 1H: The bars are too small compared to 1b and 1E. Same pattern need to be followed”.
Authors’ response: We acknowledge the Reviewer for this observation. We have modified Figure 1 to homogenise as much as possible the size of bars.
Q 4 - Specific comment: “Fig 2 & 3: It is better to make their appearance as comparative one so that the difference is visualized from their comparison”.
Authors’ response: Figures 3 and 4 contain more information than Figure 5 because we analysed 8 neuronal compartments in pyramidal ells, whereas we analysed 3 in granule cells. Nevertheless, we composed a new Figure following this suggestion and the suggestion of Q5. The resulting Figure is too large. Therefore, we have addressed this query putting together Figure 2 and 3 in a new Figure 2, but Figure 4 (now, Figure 3 in the revised version of this manuscript) must stand alone.
Q 5 - Specific comment: “Fig 4: Information pertaining to Fig 4 should follow the combination of fig 2 & 3 in a similar manner as fig 5”.
Authors’ response: Please see response in Q4.
Q 5 - Specific comment: “The sections need to be revised as per the revised figures”.
Authors’ response: Following the suggestion of the Reviewer, we have revised the sections after re-numbering some Figures.
Q 6 - Specific comment: “Section conclusion: A short section highlighting importance of the study and future directions with possible limitations would add strength to the study undertaken”.
Authors’ response: Following the suggestion of the Reviewer, we have added a short paragraph containing highlighting importance of the study and future directions.
Round 2
Reviewer 2 Report
Authors have successfully answers all my queries. The manuscript can be accepted for publication in current form.